# Time-lapse confocal imaging helps to reveal a secret behind gynoecium development

Wiktoria Wodniok [ID]

Institute of Biology, Biotechnology and Environmental Protection, Faculty of Natural Sciences, University of Silesia, Katowice, Poland

## Insights

confocal microscopy; gynoecium; MorphoGraphX; time-lapse imaging.

**Corresponding author:**
Wiktoria Wodniok;
Email: wiktoria.wodniok@us.edu.pl

**Associate Editor:**
Dr. Ali Ferjani

## Abstract

Organ morphogenesis is a complex process and numerous factors must be considered while choosing a method for its quantitative investigation. Few methods facilitate *in vivo* imaging. These are sequential replica methods combined with scanning electron microscopy and sequential confocal microscopy imaging. The latter is now the most used method to study spatiotemporal changes of organ geometry, growth and involvement of molecular factors in regulating organ development. The time-lapse confocal imaging combined with quantitative analysis of the spatiotemporal pattern of auxin efflux proteins (PIN-FORMED) was used to investigate growth and morphogenesis of *Arabidopsis* gynoecium and enabled detailed insight into gynoecium development. Yet time-lapse imaging of the gynoecium, concealed within a flower bud, presents challenges in ensuring high-quality data during all the stages of such investigations (sample preparation, maintenance of growing organ during the relatively long time of its development, laser exposure time, etc.). Analysis of vast quantitative data was facilitated by MorphoGraphX.

Morphogenesis is a highly controlled process of forming an organism through developing tissues and organs of a particular shape (Sampathkumar, 2020). Plant organ development starts at meristems, where the so-called initial cells (analogues of animal stem cells) are located. Divisions of initial cells are asymmetric in terms of daughter cell identity: one of the daughter cells and its progeny will all differentiate while the other will preserve initial cell identity. Thus, the initial cells always remain in the meristem and give rise to all the new organs. During the transition from the vegetative to reproductive phase of development, the shoot apical meristem transforms into the inflorescence meristem (or less often into the flower meristem), which will give rise to flowers. Flower development starts with the formation of flower primordium. While the flower primordium develops sepal primordia are first to be initiated but soon they enclose the flower primordium. Stamen and petal primordia begin to emerge next (in *Arabidopsis*, stamens are initiated before petals). At this stage of flower development, sepals cover the whole flower primordium. Finally, a future gynoecium starts to appear at the centre of the bud (Smyth et al., 1990). Gynoecium is the female part of the Angiosperm flower. It is made up of free or fused carpels, which are considered to be modified leaves, and consists of the ovary, the style and the stigma. In *Arabidopsis*, the gynoecium is made up of two fused carpels (Herrera-Ubaldo & de Folter, 2022). During its development, the walls of the carpels form the lateral domains and become the valves; after pollination, they will be attached to the replum (Herrera-Ubaldo & de Folter, 2022). The marginal meristem of carpel gives rise to the placenta, ovules, septum and transmitting tract. The gynoecium, which develops as an open tube, at this stage undergoes closure at the tip, which will give rise to the style and the stigma during later stages of development.

Studying plant morphogenesis can be easier compared to animals due to the lack of cell migration that enables reconstruction of cell divisions and expansion based on one-time observation of cell wall pattern (Crawford et al., 2004; Hejnowicz & Włoch, 1979; Silk et al., 1989; Zagórska-Marek & Turzańska, 2000). Such clonal analysis is straightforward as long as the organ growth is steady, like in root apex. Otherwise, time-lapse observations (sequential *in vivo* imaging) of the cell wall pattern are necessary. Studies of organs development such as

**Table 1.** Comparison of methods used for imaging plant organ development

| Methods | Sample fixation requirement | Visualised object | Sample damage | Inner cells visualised | Signal from molecular markers |
|---|---|---|---|---|---|
| One-time light microscope | No | Fixed or live sample; single stage of organ development | If fixed – sample killed during fixation | Yes | No |
| One-time SEM imaging | Yes | Fixed sample; single stage of organ development | Sample killed during fixation | No | No |
| Sequential replicas | No | Epoxy resin casts from organ surface moulds; consecutive stages of individual organ development | No need for organ isolation but damage likely during mould taking | No | No |
| Time-lapse imaging in CSLM, MPM, SRM and LSFM | No | Observation conducted on one isolated organ/sample for a certain amount of time | Organ isolation required but the sample can be grown on medium for many days | Yes | Possible |

gynoecium, which is hidden in the flower bud, are technically challenging. For this reason, gynoecium development was long veiled in mystery. Choosing the proper method to study such complex processes in hard-to-access organs is crucial. Different sequential *in vivo* imaging techniques are used for studying plant development (Table 1). One of those techniques uses the scanning electron microscope (SEM). The SEM used in direct way requires special sample preparation and fixation, which excludes sequential imaging. This obstacle has been overcome by a replica method (Green et al., 1991; Hernández et al., 1991., Kwiatkowska & Dumais, 2003, Kwiatkowska & Burian, 2014). In this technique, a silicone dental impression polymer is applied onto the growing tissue (e.g. shoot apex, young leaf) in order to obtain a mould of the sample surface. Epoxy resin casts made using the mould can then be trimmed and observed using the SEM after sputter coating. The size (dimensions) of an analysed sample is limited only by the SEM technical parameters (image aberrations at minimum magnification, chamber size, etc.). The advantage of replica method is that it eliminates the need for labelling; however, it only allows for the observation of the organ's surface. The replica technique was used to follow surface growth of an individual plant organ for up to 15 days (one replica per day) without any damage observed (Green et al., 1991). However, modern studies on plant development require measurements of not only cell divisions and growth but also gene expression levels, using, for example, reporter lines. This creates the need for imaging methods that enable tracking of cell wall pattern *in vivo* (without fixation) while observing expression of reporter genes in the same tissue. Various types of microscopes have been used for *in vivo* observation such as the confocal scanning laser microscope (CSLM), multiphoton microscope (MPM), light-sheet fluorescence microscope (LSFM) and super-resolution microscope (SRM). The CSLM is widely used in plant biology (Hoermayer et al., 2024; Li, Jenke, et al., 2024a; Mollier et al., 2023; Skrzydeł et al., 2021; Vijayan et al., 2021) since it enables deep sample penetration using a laser beam, as well as optical sectioning for 3D reconstructions of imaged samples (Elliott, 2020). CSLM is an important tool for *in vivo* imaging, including observation of spatiotemporal changes using a time-lapse technique. It is an excellent tool for observation and imaging at the cellular scale. For subcellular scale imaging, SRM may be a better choice due to its high resolution. The SRM has been used for imaging organelles' and microtubules' organization within cells (Higa et al., 2024; Li, Moreau, et al., 2024b; Molines et al., 2018; Ovečka et al., 2022; Vavrdová et al., 2020).

One of the limiting factors of the CSLM is the sample size and thickness. This obstacle has been overcome by LSFM which enables observation of samples ranging in size from tens of micrometers to several millimeters (Ovečka et al., 2018). The LSFM has gained popularity in plant research (Capua & Eshed, 2017; Vyplelová et al., 2017) not only because it can be used for observations of larger samples but also because it causes less sample damage. The LSFM, like CSLM, has been utilised for sequential *in vivo* imaging (Valuchova et al., 2020). The MPM in turn uses near-infrared pulsed lasers, which like in the case of CSLM, facilitate penetration of the sample with an advantage of lower phototoxicity. For this reason, it is a useful tool for *in vivo* imaging of various biological samples (Czymmek et al., 2007; Giri et al., 2018; Gooh et al., 2015; Kimata et al., 2016).

To summarize, a number of factors need to be considered while choosing the proper method to study complex processes such as organ growth and morphogenesis. One should select the type of microscope preferable for the study (SEM, CSLM, MPM, LSFM or SRM) based on the organ size and accessibility as well as on the required and available labelling (cell walls, reporter genes, etc.). As an example of plant development study using cutting-edge imaging and computational methods, below I am presenting a study by Gómez-Felipe et al. (2024), which focuses on analysing gynoecium growth and cell differentiation using *in vivo* time-lapse imaging. I will use the opportunity to discuss the various technical difficulties that may be encountered while conducting this type of study, related in particular to: sample preparation; sample growth condition between imaging; time-lapse imaging; and imaging data analysis (tools for confocal microscope z-stack data analysis).

In the study by Gómez-Felipe et al. (2024), inflorescences from 3-week-old *Arabidopsis* plants were isolated. To uncover gynoecium, the oldest floral buds, as well as sepals, petals and stamens already initiated on the flower bud of interest, were removed using fine tweezers and needles under a stereoscopic microscope. Gynoecium development was imaged *in vivo* for up to 13 days. Hence, it was crucial not to damage future gynoecium cells during the isolation process. This is especially problematic when working on the model plant *Arabidopsis*, the flowers of which are very small. Such isolation, therefore, requires very precise tools and operation since the slightest misstep can lead to tissue damage. In the case of time-lapse imaging, this is crucial, as we want the object of study to survive as long as possible and under the least stressful conditions possible. All this added together makes the sample preparation process challenging and time-consuming.

As mentioned before, Gómez-Felipe et al. (2024) conducted observation for 13 days, which is a long time for time-lapse

imaging, especially taking into account that the sample has been isolated from the rest of the plant and has been imaged at 24-hour intervals. To keep isolated plant organs alive for such a long time period in stressful conditions, selecting a suitable medium and the right growth conditions is critical. Isolated organ needs not only to survive between imaging but also to keep growing in a manner as similar to intact organs as possible. In the discussed paper, plants were kept in a growth chamber under long-day conditions, which are preferable conditions for *Arabidopsis* to flower.

While imaging, it is essential to choose the correct parameters of the laser, such that, on the one hand, they provide a good image quality and high enough resolution while, on the other hand, do not lead to tissue damage. Hence, laser power and exposure time need to be adjusted appropriately. Another difficulty is the proper positioning of the sample in the medium so that it is well exposed, does not move, and is not damaged in any way by the placement. Gómez-Felipe et al. (2024) placed an isolated inflorescence on a petri dish with agar medium in which a cavity had been made so that the medium immobilises the sample. Imaging the sample using a long-distance working objective makes moving of the sample unnecessary (any movement of the sample poses a risk of damage). Additionally, the horizontal placement of an inflorescence enables imaging of only one (chosen beforehand) side of the sample which ensures that the same portion of the sample is imaged every time (Silveira et al., 2022).

Even though time-lapse confocal imaging is an excellent method for studying plant development, the amount of data obtained can be overwhelming. There are various commercially available programs for confocal microscope z-stacks analysis, such as Amira (Thermo Fisher Scientific), AIVIA (Leica Microsystems) or Imaris (Oxford Instruments). Amira is a software which allows for 3D reconstructions and measurements (cell area, length, volume, sphericity) of organelles and organ structure from SEM images, as well as confocal z-stacks, and has been used previously in plant biology (Moreno et al., 2006). AIVIA enables visualisation and analysis of images and z-stacks from 2D to 5D using machine learning as one of its tools. Imaris is a software dedicated to the analysis of 3D images. It enables visualisation of cell organelles and calculation of volume, area or sphericity. One of the most popular free-of-charge software is Fiji (ImageJ), which enables image processing (e.g. geometry operations, colour processing) and analyses such as measuring area, length or stack processing. All of these software packages are useful tools for visualisation and quantitative analysis of confocal z-stacks but none is dedicated to growth quantification. However, current studies on organ development require complex analyses of temporal and spatial interplay between tissue and cell growth, morphogenesis, and gene/protein expression. The perfect software for this is a freely available MorphoGraphX that enables analysis and, more importantly, quantification of various growth and geometry parameters from confocal image stacks together with gene/protein expression patterns. The main advantage of MorphoGraphX is that the majority of analyses are performed on a curved (2.5D) surface and can be conducted over multiple time points (Barbier de Reuille et al., 2015) while 3D reconstruction of cell volume is also enabled.

Gómez-Felipe et al. (2024) used MorphoGraphX for the quantification of growth rates and anisotropy (principal growth rates and directions), cell divisions and geometry. They also employed tools for analysis of growth rates along longitudinal and mediolateral axes of the developing organ, reverse lineage tracking, and for creating heatmaps of PIN protein localization in the membrane.

Calculating growth rates along both axes was conducted using one of the newest functions of the software – the Bezier grid (Figure 1). The Bezier grid, placed on a mesh surface, allows aligning cell axis calculations in a specific direction. This method enabled authors to conclude that gynoecium growth is controlled by two orthogonal gradients (longitudinal and mediolateral).

One of the key functions of MorphoGraphX, as a tool for time-lapse image analysis, facilitates easy recognition of cell lineage (by assigning "parents"). On this basis, correspondence between data from different time points is obtained and clones derived from the cells present at the beginning of observations are identified for consecutive time points. This allows for the tracking of changes of various parameters from one time point to another in the same group of cells. The *parent* function can be used for lineage tracking analysis. This allowed the tracking of cells from different regions of an organ (gynoecium, stamen) by computing corresponding cell lineages over multiple time points (Gómez-Felipe et al., 2024; Silveira et al., 2022). The origin of the different regions of the gynoecium was characterised using lineage tracking and by finding points in the early stages of the organ development where particular groups of cells contributed exclusively to the development of a given region (Silveira et al., 2022). In the case of Gómez-Felipe et al. (2024) study, lineage tracking was used to track cells from different parts of the gynoecium, such as replum, valves and style. Cell lineage tracking is a practical and widely used tool in plant development biology (Burian et al., 2016; Kierzkowski et al., 2019; Mollier et al., 2023; Silveira et al., 2022; Zhang et al., 2020). Additionally, the software facilitates the generation of heatmaps based on the calculated data obtained from the interval between the two time points (like various growth parameters) or for the individual time-points (e.g. local curvature or intensity of reporter gene signal), which helps visualise results in a clear and accessible way.

MorphoGraphX allows the user also to overlap images from two different channels. This function helps track fluorescent landmarks (Elsner et al., 2018) or fluorescent protein expression produced in the analysed portion of a sample. By overlapping two confocal images, the user can analyse data on cell expansion and protein expression by combining geometry/growth maps with fluorescent signals. Gómez-Felipe et al. (2024) used the image overlapping to show PIN localization in plasma membranes and to create heatmaps of signal intensity. To determine auxin's role in controlling orthogonal gradients of gynoecium differentiation, they used reporter lines to visualise response to auxin and compared expression of PIN proteins with growth and differentiation of style cells. This observation proved that auxin is responsible for the longitudinal gradient in the style. To test a hypothesis that the second (mediolateral) gradient is specific to the valve, Gómez-Felipe et al. (2024) inhibited valve formation using NPA. After NPA treatment, plants were devoid of valves and showed lack of mediolateral gradient. Gómez-Felipe et al. (2024) created heatmaps showing PIN protein signal intensity by using one of the MorphoGraphX functions (*signal*) which facilitates quantification of an average signal intensity per cell. Such signals represented local auxin concentration and biosynthesis.

To sum up, Gómez-Felipe et al. (2024) used *in vivo* imaging of high spatial and temporal resolutions and sophisticated computer analysis for identification of growth and differentiation gradients that were not yet reported for *Arabidopsis* gynoecium. These gradients can be interpreted as a modification of gradient-based control of leaf morphogenesis. Auxin is putatively involved in establishment of one of the gradients.

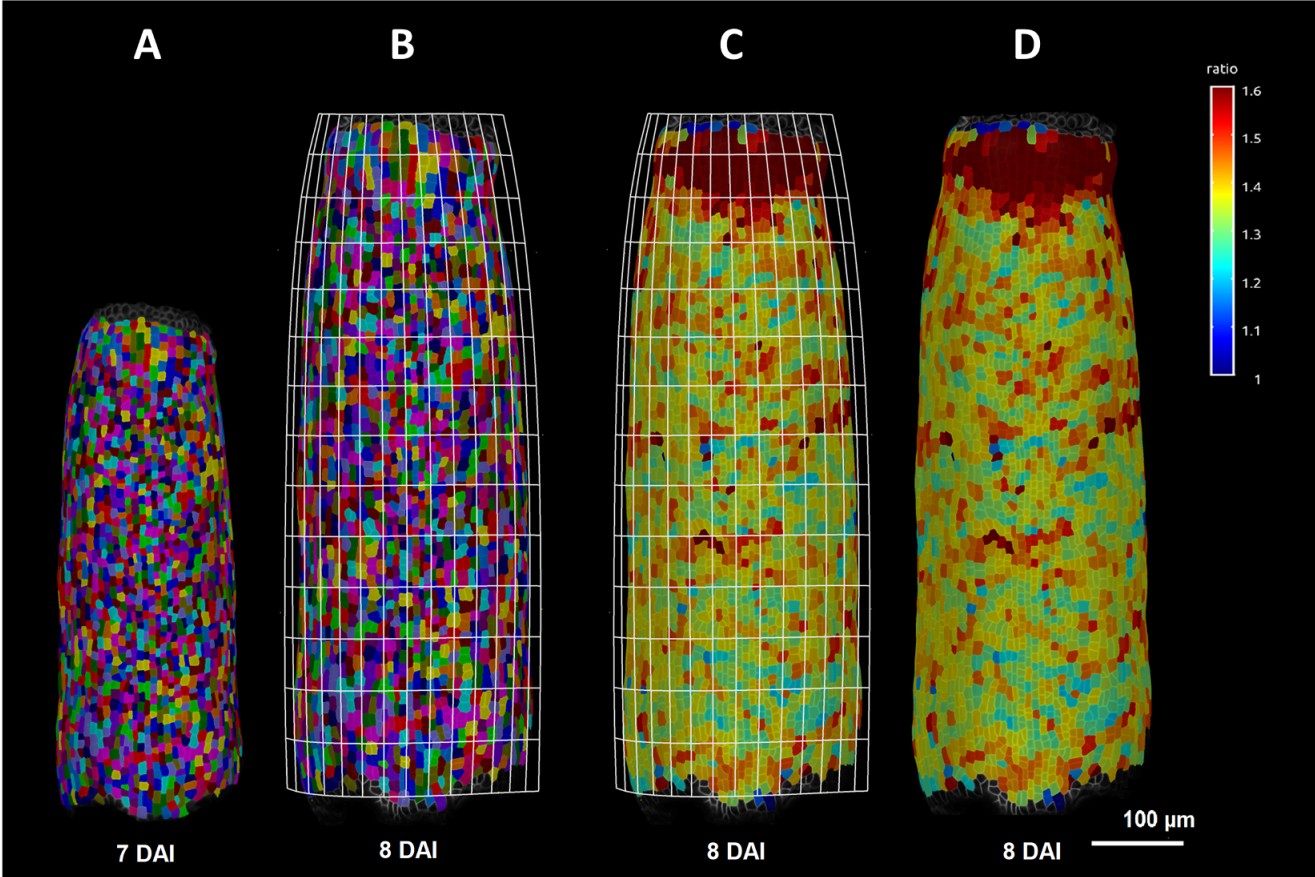

**Figure 1.** Segmented mesh of gynoecium 7 DAI (a) and after 8 DAI (b) with Bezier grid overlaid on its surface. Bezier grid enabled quantification of the cell growth rate in the area along the gynoecium axis (c). Growth heat map obtained using the Bezier grid is shown in (d).
*Source*: Images kindly provided by Daniel Kierzkowski.

It is noteworthy that the time-frame of *in vivo* imaging performed by Gómez-Felipe et al. (2024) was such wide that the first gynoecium stacks consisted of approximately 70 cells, whereas at the last time point, there were around 11000 cells. Manual processing of such an amount of data would obviously be impossible and thus a software like MorphoGraphX, which allows processing of huge amounts of data for time-lapse imaging, was crucial for such analysis.

**Open peer review.** To view the open peer review materials for this article, please visit http://doi.org/10.1017/qpb.2025.10009.

**Data availability statement.** No data or code were developed for this manuscript.

## Acknowledgements

I thank Prof. Dorota Kwiatkowska for the critical reading of the manuscript and Dr. Daniel Kierzkowski for sharing unpublished images of *Arabidopsis* gynoecium.

**Author contributions.** This manuscript was conceived and written by W.W.

**Funding statement.** This work is supported by research grant OPUS24 2022/47/B/NZ3/01972 from the Polish National Science Centre.

**Competing interest.** The author declares none.

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
