## [Reviewer Report]

The manuscript by Wiktoria Wodniok provides an overview of time-lapse imaging techniques applied to Arabidopsis gynoecium development, specifically focusing on the study by Gómez et al. (2024). The manuscript effectively summarizes the imaging and analytical techniques used and addresses practical challenges associated with these methods. This work holds potential value for advancing understanding of image analysis in plant morphogenesis and may contribute insights within the field of quantitative plant biology. However, I believe substantial revisions are required before the manuscript can be considered for publication. My main comments are listed below.

Major Points:

1. Alignment of title, abstract, and content

The title and abstract of this manuscript suggest a comprehensive discussion of time-lapse imaging methodologies and challenges across plant morphogenesis studies. However, the main text predominantly focuses on Arabidopsis gynoecium development as investigated in Gómez et al. (2024), with limited discussion of other methodologies and studies. While I understand that a comprehensive review may not be necessary under the “Insights” category, I recommend revising the title and abstract to better align with the manuscript’s content. Specifically, a title and abstract that reflect the primary focus on Gómez et al. (2024) would more accurately convey the scope to readers.

2. Clarity and appropriateness of Table 1

Table 1 presents information on “Sample size,” but there is no corresponding explanation in the main text. Information included in the table should be supplemented within the text to prevent ambiguity, particularly regarding “Sample size.” Additionally, “Cell wall labeling requirement” implies a universal need for cell wall labeling, which may not always be necessary and depends on the study’s objective. To avoid misinterpretation, I suggest adding explanations for “Sample size” in the main text and reconsidering the appropriateness of including “Cell wall labeling requirement” as a fixed category.

3. Usefulness and content of Figure 1

Figure 1 visually represents different imaging methods used in plant morphogenesis studies (Replica method, Confocal time-lapse imaging, SEM imaging). However, the figure lacks a detailed legend, and the specific purpose and advantages of each technique are not clearly conveyed. Additionally, the content may be too basic for plant cell biology specialists and does not provide new insights. I suggest replacing it with a figure that demonstrates specific examples of visualization or quantitative analysis using software, which could offer readers a fresh perspective.

Furthermore, there is no need to restrict the discussion to confocal time-lapse imaging, as other techniques, such as light-sheet microscopy and conventional wide-field fluorescence microscopy, may be more appropriate depending on the materials and objectives. Examples of analyses using these alternative methods exist, and expanding the discussion to consider other imaging techniques in both the text and figure would provide a broader view and deepen the discussion.

4. Description of MorphoGraphX and alternative software

The manuscript describes MorphoGraphX as a “perfect” software solution for image analysis but lacks mention of alternative software or approaches. An over-reliance on a single software may affect the balance of the review. Additionally, other tools may be more suitable depending on the analysis purpose or data characteristics. For example, Fiji, Imaris, Amira, and AIVIA are also available and commonly used in plant morphogenesis studies. A brief comparison of these tools’ strengths and applicability would help readers make informed choices about software selection.

Minor Point:

In line 159, the year of publication for Silveira et al. is incorrectly listed as “2021” and should be corrected to “2022.”

---

## [Reviewer Report]

This paper, entitled “Time-lapse imaging of plant development - application and challenges”, discusses the effectiveness and difficulties of time-lapse observation using confocal microscopy in plant developmental biology. This paper focuses on the recent research of Gomez et al. (2024), especially on the technical aspect of long time-lapse imaging of the gynoecium and the analysis using the MorphoGraphX software. Overall, the contents of the paper is very limited to the research of Gomez et al. (2024) and the title and abstract are obviously inadequate. The analysis of plant tissues with MorphoGraphX is already well known in the field, although Gomez et al. (2024) used the latest version of the software. It is true that the research of Gomez et al. is excellent. However, in recent years, many technical challenges have been reported in plant biology, including multiphoton microscopy, light-sheet microscopy, super-resolution microscopy, and automatic tracking using artificial intelligence. This paper does not present any of these challenges. The scope of this paper is too narrow to provide insights into time-lapse imaging of plant development. The present manuscript is rather a commentary on the work of Gomez et al. and would be better submitted in that category after rewriting the title and abstract to reflect the main contents of the paper.

---

## [Editor Report]

Dear authors,

In this manuscript, Wiktoria Wodniok provides an overview of time-lapse imaging techniques applied to Arabidopsis gynoecium development, by specifically focusing on the study by Gómez et al. (2024). The manuscript effectively summarizes the imaging and analytical techniques used and addresses practical challenges associated with these methods. 

Now we have received the comments of two reviewers experts in the field. They found that while this manuscript holds potential value for advancing understanding of image analysis in plant morphogenesis and may contribute insights within the field of quantitative plant biology, they also found that certain aspects require further attention. Therefore a substantial revision is required before the manuscript can be considered for publication in QPB. Their major comments are 

(1). Alignment of title, abstract, and manuscript contents.

(2). Clarity and appropriateness of Table 1. 

(3). Usefulness and content of Figure 1. 

(4). Description of MorphoGraphX and alternative software.

More specifically, although this paper, discusses the effectiveness and difficulties of time-lapse observation using confocal microscopy in plant developmental biology, it focuses only on the recent research of Gomez et al. (2024), especially on the technical aspect of long time-lapse imaging of the gynoecium and the analysis using the MorphoGraphX software. Overall, the contents of the paper is very limited to the research of Gomez et al. (2024) which makes the title and abstract obviously inadequate. 

The analysis of plant tissues with MorphoGraphX is already well known in the field, although Gomez et al. (2024) used the latest version of the software. It is true that the research of Gomez et al. is excellent. However, in recent years, many technical challenges have been reported in plant biology, including multiphoton microscopy, light-sheet microscopy, super-resolution microscopy, and automatic tracking using artificial intelligence. Unfortunately, this paper does not present any of these challenges. 

The scope of this paper is too narrow to provide insights into time-lapse imaging of plant development in its wider context. The present manuscript is rather a commentary on the work of Gomez et al. and would be better submitted in “Insights” category after rewriting the title and abstract to reflect the main contents of the paper.

As you may have appreciated , the reviewers have raised critical points about the whole manuscript, the most important of it is its narrow scope. As a minimum requirement the authors are invited to extensively revise the manuscript based on the suggestion of both reviewers.

Thank you again for submitting your work to QPB, and I am looking forward to receiving your revised manuscript.

Ali FERJANI

Associate Editor of QPB

---

## [Reviewer Report]

I have carefully read the revised manuscript by Wiktoria Wodniok, which I previously reviewed.

I appreciate the authors’ thorough revisions in response to my previous comments. The manuscript has significantly improved in clarity and depth, and I am confident that its quality as a review article has been enhanced.

The expanded discussion on software tools has made the manuscript more balanced and comprehensive, offering a fairer perspective on the topic. Additionally, the newly added Figure 1 is not only visually appealing but also highly insightful, effectively conveying key concepts in an intuitive manner.

In light of these improvements, I fully support the publication of this review article.

---

## [Reviewer Report]

This paper describes insights into the effectiveness of long-term time-lapse microscopy on plant organ development. In this paper, the author first provides general information about microscopy in plant developmental biology and then focuses on the recently published paper by Gomez-Felipe et al. (2024), which describes the long-term observation of gynoecium development by CLSM and the computational analysis of the data using MorphoGraphX. I think the manuscript is well written and provides a good introduction to the long-term observation approach to plant organ development. I have only a few minor suggestions as listed below.

1. line #110, light-sheet microscope (LSM) should be light-sheet fluorescence microscope (LSFM).

2. Line#118, there are more examples of super-resolution microscopy in plants, such as Ovečka et al. 2022 Plant Physiol (review), Higa et al. 2024 Nature Plants, Ziqiang Li et al. 2024 Science, etc, some of which could be included in the reference.

3. Line#128, research on Arabidopsis zygote could be a good example of MPM, such as Kimata et al. 2016 PNAS and Gooh et al. 2015 Dev Cell.

4. Table I should be reorganized. EM and light microscopy should be separated into different rows. One-time light microscopy is not a classification of microscopy. I suggest the authors compare the techniques of microscopy such as CLSM, CLFM, MPM, SRM, and SEM for a more comprehensive understanding of microscopy in plant developmental biology. Signal of molecular markers in SEM imaging is no?

---

## [Editor Report]

Dear authors,

Now we have received the comments from the reviewers on he manuscript QPB-2024-0052.R1. 

I really appreciate the authors’ thorough revisions in response to the reviewer’s suggestions and comments. The reviewers found that the manuscript has significantly improved in clarity and depth, and I am confident that its quality as a review article has been enhanced too.

The expanded discussion on software tools has made the manuscript more balanced and comprehensive, offering a fairer perspective on the topic. Additionally, the newly added Figure 1 is not only visually appealing but also highly insightful, effectively conveying key concepts in an intuitive manner. However, one of the two reviewers has suggested minor issues to e considered before formal Acceptance of the manuscript (please see his/her minor comments below).

In the light of the above, I am happy to fully support the publication of this review article once the minor issues have been addressed.

---

## [Reviewer Report]

I am satisfied with the authors’ responses and the improvements made. I have no further comments and recommend the manuscript for publication.

---

## [Reviewer Report]

I found that the author appropriately revised the manuscript. But I just wonder that, in table 1, “signal from molecular markers” of one-time light microscope might be “possible”, if it includes fluorescence light microscopy.

---

## [Editor Report]

Dear Dr. Wodniok Wiktoria,

Thank you for resubmitting your revised manuscript. Based on the reviewer’s feedback and on my own evaluation I am happy to recommend your insights paper for publication in QPB. 

Thank you again for submitting your nice work to QPB

Ali FERJANI